# Bilateral Condylar Hyperplasia: Importance of Its Diagnosis in the Treatment and Long-Term Stability of Skeletal Class III Correction

**DOI:** 10.3390/diagnostics15070809

**Published:** 2025-03-22

**Authors:** Diego Fernando López, Martín Fernando Orozco, Sofia Ochoa Gómez, Santiago Herrera Guardiola, Luis Eduardo Almeida

**Affiliations:** 1Orthodontics Department, Universidad del Valle, Cali 760043, Colombia; santiago.ortodonciaestetica@gmail.com; 2Department of Maxillofacial Surgery, Universidad del Bosque, Bogotá 110111, Colombia; martinorozcofernandez@gmail.com; 3Department of Orthodontics, UniCIEO, Bogotá 110111, Colombia; sofiochoagomez@gmail.com; 4Surgical Sciences Department, School of Dentistry, Marquette University, Milwakee, WI 53233, USA; luiseduardoalmeida@marquette.edu

**Keywords:** hyperplasia, mandibular condyle, radioisotope scanning, diagnostic imaging, mandibular prognathism

## Abstract

**Background:** Condylar hyperplasia (CH) leads to mandibular overgrowth with anatomical, aesthetic, and functional consequences, particularly affecting facial harmony. It is characterized by severe mandibular prognathism (MP) in bilateral cases. This study aims to propose a therapeutic algorithm for diagnosing and treating bilateral condylar hyperplasia (BCH) based on demographic, clinical, craniofacial growth, and clivus ratio uptake conditions. **Methods:** Ten patients with severe skeletal Class III by MP, whose alteration was clinically associated with BCH, were consecutively evaluated in a specialized dentofacial deformity center between the period of 2019 and 2024. A detailed protocol was followed to gather clinical history, assess anatomical features, evaluate malocclusion, and identify potential BCH. When suspicion arose, a nuclear medicine test measured condylar scintigraphy uptake. If the result was positive, patients underwent bilateral condylectomy, following one of three treatment protocols. **Results:** Severe PM, pronounced Class III with excessive negative overjet, elongated condyles of normal anatomy, absence of family history, and accelerated growth since preadolescence and adolescence were common characteristics in these patients. Regarding the treatment protocol chosen according to the characteristics of the patients, five cases followed treatment protocol A: condylectomy and surgical correction of the alteration in two surgical stages. Two cases followed protocol B: bilateral condylectomy and orthognathic surgery in the same surgical time, and three cases followed protocol C: condylectomy and later post-surgical orthopedics and/or orthodontics without a second surgical intervention. Histopathological results confirmed bilateral hyperplastic growth and stability in mandibular size, and occlusion was observed during follow-up. **Conclusions:** Specialists need to recognize the clinical signs of BCH and use scintigraphy tests to measure condylar metabolic activity when suspected. Early detection of BCH is crucial, as it influences treatment decisions and helps prevent relapses in orthodontic or surgical interventions aimed solely at correcting or compensating for Class III malocclusion caused by MP.

## 1. Introduction

Condylar hyperplasia (CH) affects facial growth due to overgrowth of the mandibular condyle, histologically characterized by an increase in the number of cells and the thickness of the layers that compose the soft condyle [1,2]. Although it is a self-limiting entity, it can be deforming because condylar overgrowth can be present during the whole stage of growth and development and even continue well beyond the residual growth [3], affecting, according to its time of evolution, type of hyperplasia, and severity, structures not only related to the temporomandibular joint (TMJ) (condyle, disc, fossa) but also to the mandibular ramus and producing in the upper jaw, secondary vertical compensations and malocclusions with sagittal, transverse, and vertical compromises [4]. Its etiology may be due to genetic, hormonal, functional, neoplastic, molecular, or traumatic factors [5].

All of the above not only leads to possible functional alterations of the TMJ [6] but also progressive facial asymmetries in the case of unilateral conditions [5,7] and severe mandibular prognathism in the case of bilateral conditions [8].

Although the diagnosis of CH and its different variations depend on a thorough and detailed correlation between extraoral and intraoral clinical features with radiographic and/or tomographic findings [4], the metabolic information of the active state of CH, which represents the subchondral bone growth activity in the mandibular condyle, can only be obtained through nuclear medicine tests, which have evolved from planar bone scans to single photon emission computed tomography (SPECT) [9] and, currently, to SPECT/CT tests, which seek to obtain combined information of the anatomical alteration together with the metabolic activity [10,11].

Regarding bilateral condylar hyperplasia (BCH), its diagnosis and, therefore, its treatment has been a challenge for clinicians because, on the one hand, there is no clarity in the anatomical characteristics of the alteration, the time of onset of the pathology, or the parameters of differentiation with skeletal Class III due to mandibular prognathism (MP), only what was proposed by Wolford et al. in 2014 [8] and the hypotheses of Goulart et al., 2015 [12] and López et al., 2024 [13]. On the other hand, few studies provide reference values for condylar metabolic activity in healthy populations that take the clivus as a reference structure for comparison with the condyle [14,15,16]. Almost all studies have focused on the percentage comparison of uptake between condyles, and this is useful only in cases of suspected unilateral CH [9].

In turn, the MP, which represents the exaggerated sagittal projection of the mandible, leads to a Class III skeletal and occlusal relationship that affects the facial profile (concave) with esthetic and functional compromise [8]. Although its etiologic factor is associated with the underdevelopment of the upper jaw, overdevelopment of the lower jaw, or a combination of both, in some cases, the excess longitudinal growth of the condyle may be the etiologic factor triggering the alteration [8].

Therefore, the aim of this article is to present a series of cases in which a clinical, anatomical, radiographic, and gammagraphic diagnostic methodology was followed for the identification of BCH according to the available evidence. A therapeutic algorithm for the clinical and metabolic diagnosis of the disorder is also presented, and three treatment protocols for BCH are proposed according to the individual characteristics of each patient.

## 2. Materials and Methods

All skeletal Class III patients for MP whose prognathism was clinically associated with BCH were retrospectively evaluated from 2019 to 2024. These patients were sent to nuclear medicine to obtain information on the bone metabolic activity of each condyle with respect to the clivus, and, if the result was positive for abnormally active condylar growth, the patients underwent bilateral condylectomy with an extraoral approach in a center specialized in the treatment of dentofacial deformities. The Declaration of Helsinki guidelines on medical protocol and ethics were followed at all stages of treatment. The conduct of this study did not alter the ethically approved protocol for the diagnosis and treatment of CH and Class III skeletal discrepancy with MP at the specialized center; therefore, it was exempt from the requirement for additional ethical approval.

Patients with any type of syndrome affecting the craniofacial configuration, as well as the mandibular fossa and the development of the middle and lower third, were excluded from this study. Also, patients with a history of craniofacial trauma or previous orthognathic surgeries. To be included, in addition to a complete clinical history, it was indispensable to have a positive SPECT test result and post-surgery histopathological result.

### 2.1. Diagnostic Evaluation and Treatment Planning

For detecting suspected bilateral condylar hyperplasia (BCH), the specialized center for dentofacial deformities follows this protocol (Figure 1):Clinical History: A detailed history is taken from the patient and family members, focusing on any family history of mandibular prognathism (MP), the onset of disproportionate growth, previous related treatments, TMJ issues, and psycho-functional concerns.Clinical Evaluation: The sagittal relationship of the bony bases, severity of Class III malocclusion, facial profile, growth type, and vertical compromise are assessed.Imaging: Two-dimensional and three-dimensional imaging are used to determine the anatomical characteristics, measure condylar length, mandibular length, and maxillary sagittal size, and compare these with population-specific reference values.SPECT/CT Scan: A SPECT/CT scan of the TMJ is performed, with radiopharmaceutical absorption ratios calculated for each condyle relative to the clivus. The formula used is as follows:$$\frac{Condylarcounts–Backgroundcounts}{Clivuscounts–Background}$$Five transaxial tomographic slices define a fixed region of interest (ROI).Surgical Planning: Surgical planning according to the information collected and the characteristics of the alteration, following three treatment schemes.
○Protocol A: Bilateral condylectomy via an extraoral approach with a modified endaural incision, followed by bracket placement and orthodontic alignment. After condylectomy, orthodontic decompensation is performed, with a second surgical phase planned using virtual surgical planning. This approach is for patients with no growth potential, severe skeletal discrepancies, and aberrant malocclusions. Post-condylectomy, clinical improvements are noted (Figure 2, Figure 3 and Figure 4).○Protocol B: Bilateral condylectomy and orthognathic surgery performed in one session, followed by post-surgical orthodontics. The surgery order is first the high Le Fort I osteotomy in bilateral step, then the bilateral condylectomies, and then the sagittal osteotomies of the bilateral mandibular ramus, ending with advancement mentoplasty when necessary. This approach is suitable for patients with no growth potential, severe skeletal issues, aberrant malocclusions, and significant psychoemotional distress due to facial appearance. It is indicated for those needing rapid correction to improve their quality of life (Figure 5 and Figure 6).○Protocol C: Bilateral high condylectomy followed by maxillary orthopedics (if growth potential exists) or compensatory orthodontics. This approach is for patients with growth potential or mild conditions, where early detection allows for correction using only condylectomy and proper orthodontics without additional surgery (Figure 7, Figure 8, Figure 9 and Figure 10).

### 2.2. Surgical Technique

The procedure is performed under general anesthesia with nasotracheal intubation. A cotton swab, impregnated with 2 mg/g Furacin ointment, is placed in the ear canal. The incision design for the modified endaural approach is marked [17], starting at the anterior third of the helix and extending to its root, passing endaurally through the internal surface of the tragus to the earlobe junction. The preauricular region is infiltrated with 1% xylocaine with epinephrine (1:200,000), 5 cm^3^, at the helix and tragus to achieve hydrodissection and vasoconstriction.

Using a #15 scalpel, the incision is parallel to the auricular cartilage, separating the skin, subcutaneous tissue, and superficial temporal fascia while protecting the facial nerve branches (frontal, zygomatic, superior buccal). The dissection proceeds with Dean scissors until the deep temporal fascia and articular capsule are reached. Lidocaine with epinephrine is used for hydrodissection, separating the capsule from the condyle. The articular disc is carefully separated from the secondary condylar cartilage with a Molt 9 dissector. TMJ separators are placed backward and outward, and the planned measurements from the tomography are marked with a caliper. Condylectomy is performed with the Mectron^®^ piezoelectric from the lateral to the medial pole.

It is decided to perform a high condylectomy when the characteristics of the alteration are not severe, generally diagnosed at an early age and in its initial stages [18]. A low or proportional condylectomy is used for severe cases where reducing the effective mandibular length is necessary [19]. The osteotomized segments are sent for pathology, and osteoplasty of the condylar stump is performed. Meniscopexy is performed from the disc to the capsule. Suturing is performed in layers using Vicryl 4/0, and the skin is closed with Prolene 6/0. A dressing with Bactigras and steri-strips is applied.

Postoperatively, the patient is prescribed cephalexin 500 mg every 8 h for 7 days and etoricoxib 120 mg once daily for 5 days. Stitches are removed seven days later, and Class III elastic vector orthodontic therapy is initiated one week postoperatively. In the third case, 250 g of orthopedic force is applied via skeletal temporary anchorage systems.

## 3. Results

From August 2019 to November 2024, 10 patients (6 males and 4 females) aged from 11 to 30 years underwent bilateral condylectomy at the study center due to progressive, exaggerated mandibular growth, resulting in severe Class III malocclusion and negative overjet. Table 1 summarizes the patients’ demographic characteristics, clinical findings, scintigraphy results, condyle removal volumes, histopathological reports, treatment scheme followed, and follow-up durations.

All patients exhibited elongated but otherwise normal condylar anatomy. The onset of disproportionate mandibular growth typically coincided with the prepubertal growth spurt in males (ages 14–16) and menarche in females. Except for case 3, who reported having a dizygotic twin with unilateral condylar hyperplasia, no family history of Class III or MP was noted. Cases 3, 4, and 6, all over 18, reported continued mandibular growth beyond the cessation of overall body growth.

The scintigraphy results revealed that, for female patients (cases 1, 2, 6, and 9), the condylar uptake ratio concerning the clivus was higher than 1.33, while male patients (cases 8 and 10) had uptake ratios exceeding 2.09. In the other cases, the nuclear medicine center report showed increased condylar uptake concerning clivus above normal age-adjusted values but did not provide quantitative data.

High condylectomy (5–6 mm condylar cut) was the predominant procedure, performed on cases 1, 2, 4, 5, 6, and 9, and for the right condyle of case 3 and the left condyle of case 7. Low condylectomy (beyond 6 mm) was performed on the left condyle of case 3 and the right condyle of case 7, and bilaterally for cases 8 and 10. The authors reported that, as more of the condylar portion was removed, the effective mandibular length decreased and, therefore, the sagittal mandibular projection diminished.

Histopathological analysis revealed that cases 1, 2, and 9 had soft condyle thicknesses exceeding 0.6 mm. The remaining patients exhibited increased proliferation of hypertrophic chondrocytes in the medullary bone, consistent with BCH.

The most common treatment approach was Protocol A (cases 1, 3, 6, 7, and 10), characterized by bilateral condylectomy, to stop the aggressive growth, then orthodontic therapy to prepare the patient for a second surgical intervention to correct the sequel according to the patient’s alterations. One of the patients who followed this protocol is the one presented in Figure 2, Figure 3 and Figure 4, which shows a 14-year-old female patient (Case 1, Table 1), with severe mandibular prognathism, that was accentuated after menarche with no family history of Class III. Anatomically, as evidenced by the radiographic images, the condyles are elongated but have normal anatomy. During the diagnostic process, a nuclear medicine test was conducted to obtain the radiopharmaceutical uptake in the condyles concerning the clivus, and the result was 1.4 for each condyle, confirming the active state of CH. With this diagnosis, it was decided to proceed with a first surgical intervention consisting of a 6 mm bilateral high condylectomy. The results of the histopathological test reported a soft condyle thickness of 0.6 mm for the right condyle and 0.7 mm for the left condyle, with positive findings for BCH. After surgery, orthodontic decompensation was performed for 12 months, and a second surgical intervention was scheduled, this time, bimaxillary to correct the skeletal alteration. Post-surgical orthodontics were then carried out for 8 months to achieve adequate occlusal stability.

Protocol C followed in frequency (cases 2, 8, and 9), a single-surgical approach, involved bilateral condylectomy followed by orthodontics and/or orthopedics to correct sagittal discrepancies, avoiding a second surgery. One of the patients who followed this protocol is the one presented in Figure 7, Figure 8, Figure 9 and Figure 10, which show a 12-year-old female patient (Case 9, Table 1) with a severe mandibular prognathism that was accentuated after menarche. Anatomically, as evidenced by radiographic images, the condyles are elongated with posterior divergence but have normal anatomy. In addition, she has a lingual offset of the lower incisors. The result of the nuclear medicine test was 1.45 for the right condyle and 1.38 for the left condyle, confirming the active state of the CH. The surgical intervention consisted of a 5 mm high bilateral condylectomy. Histopathologic examination reported a soft condyle thickness of 0.6 mm for both condyles, with positive findings for CHB. In this case, in which the sequelae in the three planes of space were not severe and, additionally, there was growth potential, it was decided to correct the sagittal relationship of the jaws with skeletally anchored orthopedics and then with corrective orthodontics to achieve an ideal occlusion. In the active phase, the treatment lasted 24 months, and, in the follow-up phase, it lasted 48 months with adequate stability.

The third scheme in frequency was Protocol B (cases 4 and 5), which involved condylectomy and orthognathic surgery in one session, followed by post-surgical orthodontics. One of the patients who followed this protocol is the one presented in Figure 5 and Figure 6, which show a 17-year-old male patient (Case 5, Table 1) with severe mandibular prognathism, accentuated since a pubertal growth spurt, and with no family history of Class III. Anatomically, as evidenced by radiographic images, the condyles were elongated with posterior divergence but presented normal anatomy. In addition, the clinical history evidenced episodes of depression due to her facial appearance and also failed orthodontic treatments to correct the skeletal alteration. The nuclear medicine center sent a positive report for bilateral hypercaptation in condyles concerning age-adjusted clivus, without providing quantitative data. In this case, because of her psychoemotional compromise, everything was performed in the same surgical time. The order of the procedure was an 8 mm high Lefort I osteotomy followed by 6 mm high bilateral condylectomies and, finally, a sagittal osteotomy of the mandibular ramus for setback, accompanied by an 8 mm advancement mentoplasty and 5 mm ascension. The histopathological findings were confirmatory of BCH and post-surgical orthodontics was performed for 24 months. The follow-up has been for 60 months, observing adequate stability in mandibular size and position, as well as in the occlusal relationship.

Follow-up durations ranged from 8 months for the most recent case to 60 months for the longest. No secondary growth was observed, and mandibular position and size remained stable post-condylectomy, as did occlusion after the completion of treatment.

## 4. Discussion

Class III skeletal malocclusion can be classified into clusters and subclusters [20]. It is characterized by the fact that about 75% of Class III patients have a skeletal component. This sagittal discrepancy of the jaws can be a consequence of maxillary underdevelopment (prevalence: 19.5%), mandibular overdevelopment (prevalence: 47.4%), or a combination of both (incidence: 8.7%) [21]. Southeast Asian populations have the highest incidence (18.5%), followed by the Middle East (10.18%) and Latin America (9.1%) [22]. The etiology is multifactorial, involving both environmental and genetic factors, and it is considered a polygenic condition influenced by ethnicity [23]. Prognosis and treatment are largely influenced by the age of onset and the severity of the condition.

BCH may be one of the etiological factors of severe MP where condylar growth, on the one hand, goes far beyond the residual mandibular growth and, on the other hand, to an exaggerated growth rate with severe esthetic, functional, and psychoemotional consequences. This continuous condylar growth could even be one of the causes of the recurrence of Class III surgical or orthodontic corrections when the diagnosis of BCH is overlooked.

Concerning BCH, there is limited literature that specifies its characteristics in detail and gives clarity to the clinician with respect to the development of the disease. For now, there are only postulations and hypotheses derived from results in morphological and metabolic studies.

Wolford et al. (2014) [8] described BCH as accelerated, disproportionate condylar growth extending into the second decade of life, resulting in Class III malocclusion with MP and anatomically elongated but normal-shaped condyles. For its treatment, they recommended bilateral condylectomy with disc repositioning and orthognathic surgery according to the patient’s age.

Regarding these anatomical changes, Mohsen et al. (2023) [24] conducted a morphological and morphometric evaluation of the mandibular condyles using CBCT, linking them to skeletal malocclusions. They found that the length of mandibular condyles in Class III malocclusions was significantly greater than in Class I or II. Similar findings were reported by Krisjane et al. (2009) [25], Saccucci et al. (2012) [26], and Noh et al. (2021) [27], all concluding that larger condylar size is a characteristic of skeletal Class III malocclusions.

Goulart et al. (2015) [12] studied a sample of 30 patients—15 with unilateral condylar hyperplasia (CH) and 15 with Class III malocclusion due to mandibular prognathism (MP). Using CBCT, they compared condylar length and the distances of the mediolateral and anteroposterior poles. Their findings showed no significant differences in condylar size between the two groups, suggesting that some cases diagnosed as Class III malocclusion could actually be cases of BCH.

Lopez et al. (2024) [13] conducted a study on volumetric differences in the temporomandibular joint (TMJ) structures—condyle, fossa, and disc—using 3D tomographic segmentation in a sample of 138 patients with facial asymmetry. Of these, 64 had hemimandibular elongation, 12 had hemimandibular hyperplasia, 20 had hybrid forms within the condylar hyperplasia (CH) group, and 25 presented with asymmetric MP. The study found that condylar volume was increased in both CH and MP cases compared to healthy controls and other conditions causing facial asymmetry. However, there were no significant volumetric differences between CH and MP, suggesting that condylar overdevelopment may be an etiological factor of MP in Class III malocclusions, as seen in BCH. This implies that many cases historically diagnosed as MP could be BCH.

Gammagraphic studies provide valuable information on bone metabolism, as radiotracers used in these studies are absorbed in areas of increased blood flow and active bone mineralization [10]. Historically, these studies have focused on comparing condylar uptake to assess active states in unilateral CH cases with facial asymmetry. Recently, efforts have been made to establish reference values for radiopharmaceutical uptake based on age and gender, using the clivus as a reference structure to compare condylar uptake [10]. This approach allows for a more individualized assessment when suspecting BCH.

In this regard, Anzola et al. conducted two studies [14,16]. In the first, involving 72 healthy patients, they found radiopharmaceutical uptake ratios concerning the clivus up to 1.28. In the second study, which included 48 healthy patients, the maximum uptake was 0.84. Lopez et al. [15], studying 80 healthy patients, found sexual dimorphism in uptake, indicating the need to differentiate values by gender. The ranges observed were from 0.46 to 1.33 for females and from 0.61 to 2.09 for males, suggesting that uptake above these ranges indicates active hyperplasia. To calculate the ratio, both condylar and clivus uptake and background radiation (near the study area) were considered.

The differences in the results, despite being studies conducted in similar populations, could be attributed to variations in methodology, such as the selection of the region of interest (ROI) for uptake extraction or the formula used. This generates confusion for clinicians because decisions regarding this pathology are related to surgical procedures and, if the best decision is not made, conditions that are deforming from the beginning could be aggravated.

To address this, the literature has proposed standardizing methodologies to ensure the test is reliable and reproducible. Therefore, our study used a fixed ROI derived from the sum of five transaxial tomographic slices based on total counts. This technique offers greater reproducibility between operators and improved intraoperator repeatability, with minimal variability and better diagnostic agreement [28]. Additionally, we applied the following formula to calculate the ratio [15]:$$\frac{Condylecounts–Backgroundcounts}{Clivuscounts–Backgroundcounts}$$

Histopathologically, CH is characterized by an increase in the thickness of the articular layer of fibrous tissue, an increase in the proliferative layer of undifferentiated mesenchymal cells, and, also, an increase in the hypertrophic cartilage layer, which together represent the soft condyle [2]. In the patients reported in this article, this thickness was increased with regard to the reference values, with some having thicknesses above 0.6 mm and also increased proliferation of hypertrophic chondrocytes on the medullary bone with a histopathological report confirming BCH.

The clinical characteristics and common background observed in these patients, as well as those treated by the dental–skeletal disorders team to which the authors belong, include the following: (1) exaggerated mandibular growth starting in preadolescence or adolescence, surpassing the typical growth rates in the population [29]; (2) progressive development of severe mandibular prognathism (MP) with increasing age; (3) severely progressive skeletal Class III malocclusion with negative overjet; (4) elongated condylar anatomy with a normal shape; (5) no family history of Class III malocclusion or prognathism; (6) mandibular growth extending beyond the typical cessation of facial growth.

The treatment approach for CH is heavily influenced by factors such as the patient’s growth potential, the timing of diagnosis, the severity of the skeletal alteration, and the presence of any psycho-functional concerns. When growth potential remains, or when the severity of the alteration is not yet fully manifested, a combination of bilateral post-condylectomy, skeletally-anchored maxillary orthopedics, and/or corrective orthodontics may be used to control the sagittal relationship of the jaws [30]. In the absence of growth potential, bilateral condylectomy and orthognathic surgery are necessary, performed in one or two stages to correct the skeletal discrepancies. The choice of treatment is often influenced by the patient’s healthcare system and insurance coverage, as well as potential mood disturbances that can affect the patient’s quality of life due to aesthetic concerns [31]. In these cases, a comprehensive approach involving condylar and maxillary surgery is typically preferred for a quicker correction of the deformity. It is important to note that, the more condyle is removed during the condylectomy, the greater the reduction in effective mandibular length and, consequently, the reduction in mandibular projection.

Although the literature has not provided data on the prevalence of BCH, nor on the natural history of the disease that characterizes this entity, nor has it standardized the diagnostic processes in it, and, although we understand that this is a novel proposal that we hope will create clinical awareness for its correct diagnosis and treatment, we understand that one of the limitations of this study is the size of the sample since it still represents a series of cases. It is necessary to carry out longitudinal studies with a larger sample size in which a comparison is made with a population of patients with severe MP not associated with BCH.

## 5. Conclusions

One important implication for clinical practice is that specialists must be aware of the existence of BCH and the clinical conditions that may indicate its presence. Scintigraphy tests can assess condylar metabolic activity when there is suspicion of this condition. BCH not only alters mandibular growth but does so disproportionately and progressively, affecting the aesthetics, function, and self-esteem of patients. Therefore, early detection is crucial, as it directly impacts treatment decisions and helps prevent relapses in orthodontic or surgical treatments solely focused on correcting or compensating for Class III malocclusion caused by mandibular prognathism. The choice of treatment protocol varies according to the patient’s growth potential, the severity of the condition, the time at which the excessive overgrowth is detected, the psycho-emotional compromise, and the magnitude of the sequelae.

## Figures and Tables

**Figure 1 diagnostics-15-00809-f001:**
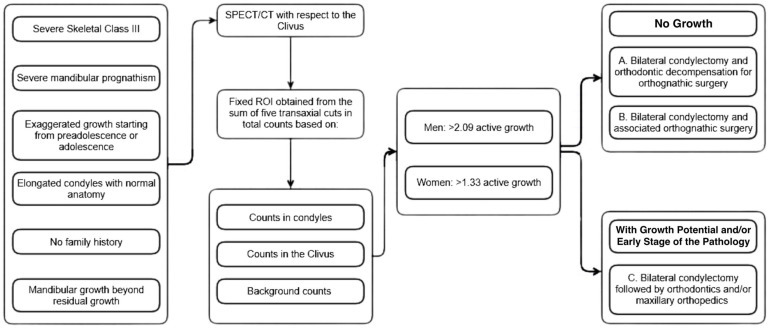
Therapeutic algorithm for identifying and treating bilateral condylar hyperplasia, considering the rate of radiopharmaceutical uptake by each condyle concerning clivus. Values > 2.09 in men and 1.33 in women are suggested to confirm active status, and three treatment protocols are presented.

**Figure 2 diagnostics-15-00809-f002:**
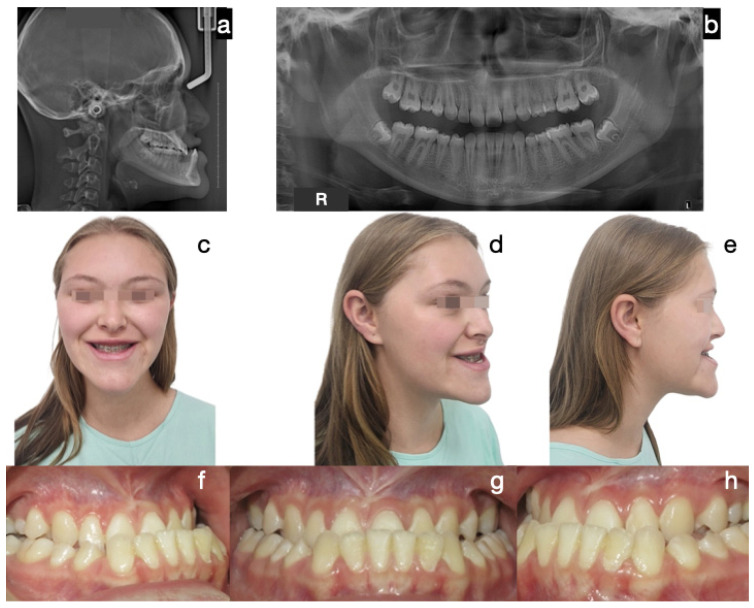
Case 1 corresponds to patient 1, described in Table 1. Protocol A (bilateral condylectomy and orthognathic surgery in two surgical stages): initial photos. The severity of the Class III skeletal and dental discrepancy with negative overjet of −12 mm and concave profile is evidenced. (**a**) Lateral skull radiograph, (**b**) panoramic radiograph, (**c**) frontal extraoral photograph, (**d**) ¾ extraoral photograph, (**e**) lateral extraoral photograph, (**f**) right lateral intraoral photograph, (**g**) frontal intraoral photograph, (**h**) left lateral intraoral photograph.

**Figure 3 diagnostics-15-00809-f003:**
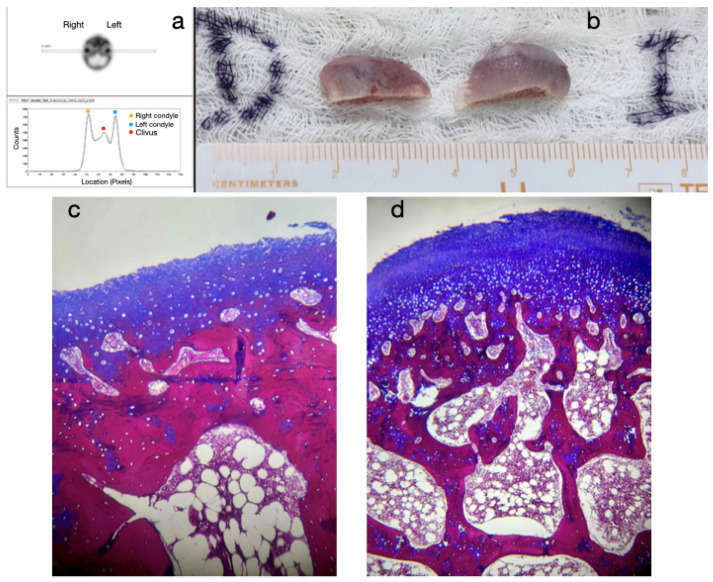
(**a**) Histogram of the SPECT test with uptake ratio for both condyles to clivus of 1.4; (**b**) bilateral high postcondylectomy specimens; (**c**) image of the right histological slice with 0.6 mm thickness in the soft condyle; (**d**) image of the left histological slice with 0.7 mm thickness in the soft condyle.

**Figure 4 diagnostics-15-00809-f004:**
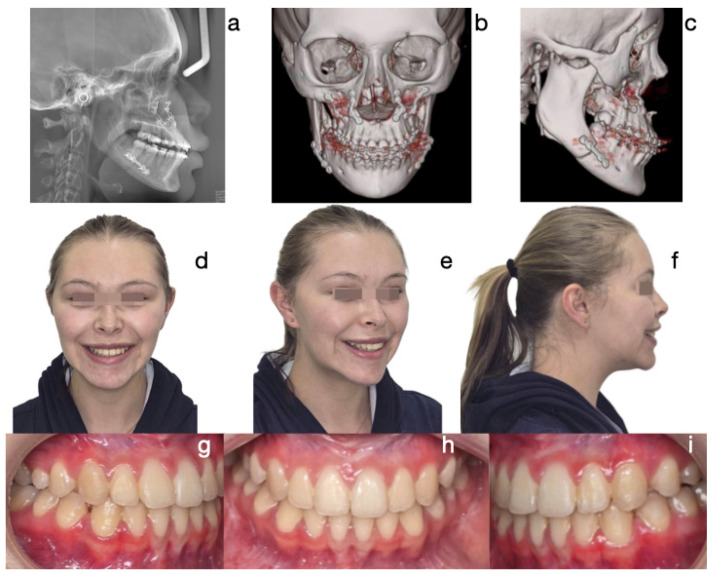
Post-surgery photographs: (**a**) lateral skull X-ray, (**b**) frontal CBCT, (**c**) lateral CBCT. Final photographs. Correction of Class III skeletal and dental discrepancy. Straight facial profile, positive overjet, and stable occlusion. (**d**) Frontal extraoral photograph, (**e**) ¾ extraoral photograph, (**f**) lateral extraoral photograph, (**g**) lateral right intraoral photograph, (**h**) frontal intraoral photograph, (**i**) lateral left intraoral photograph.

**Figure 5 diagnostics-15-00809-f005:**
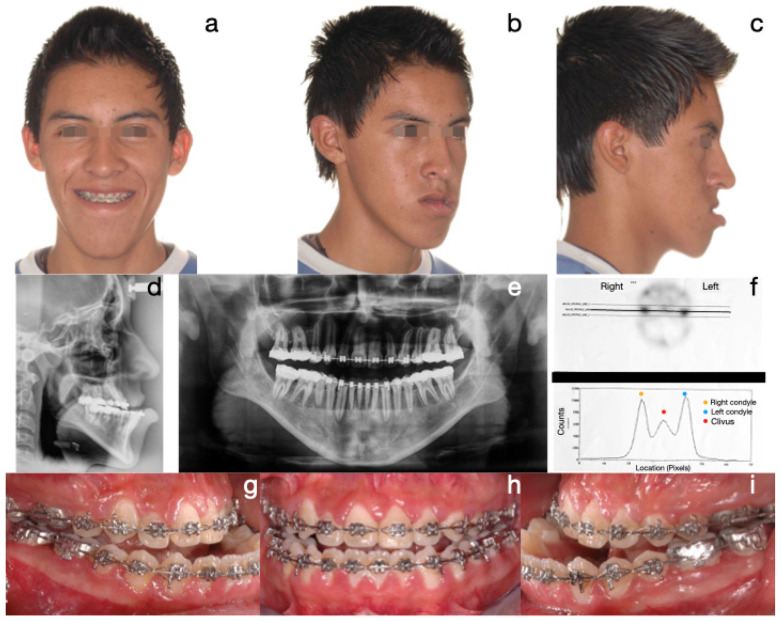
Case 2 corresponds to patient 5, described in Table 1. Protocol B (condylectomy and orthognathic surgery simultaneously): pre-surgical photos. The severity of the Class III skeletal and dental discrepancy with negative overjet of −16 mm and concave profile is evidenced. (**a**) Frontal extraoral photograph, (**b**) ¾ extraoral photograph, (**c**) lateral extraoral photograph, (**d**) lateral skull radiograph, (**e**) panoramic radiograph, (**f**) histogram of the SPECT test with uptake ratio above normal values to clivus. Pre-surgical photos: (**g**) intraoral right lateral photograph, (**h**) intraoral frontal photograph, (**i**) intraoral left lateral photograph.

**Figure 6 diagnostics-15-00809-f006:**
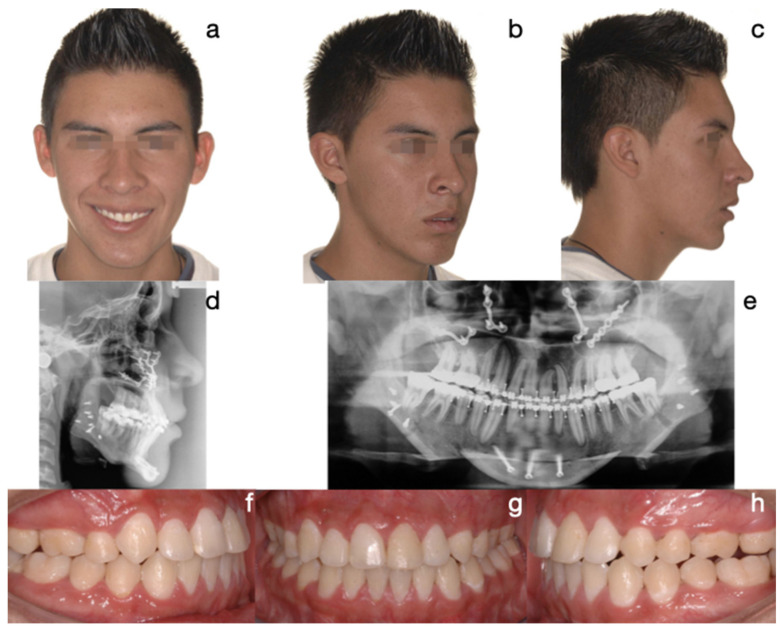
Final photos. Correction of skeletal discrepancy, positive overjet, improvement of profile, and malocclusion. (**a**) Extraoral frontal photograph, (**b**) ¾ extraoral photograph, (**c**) extraoral lateral photograph, (**d**) lateral skull radiograph, (**e**) panoramic radiograph, (**f**) intraoral right lateral photograph, (**g**) intraoral frontal photograph, (**h**) intraoral left lateral photograph.

**Figure 7 diagnostics-15-00809-f007:**
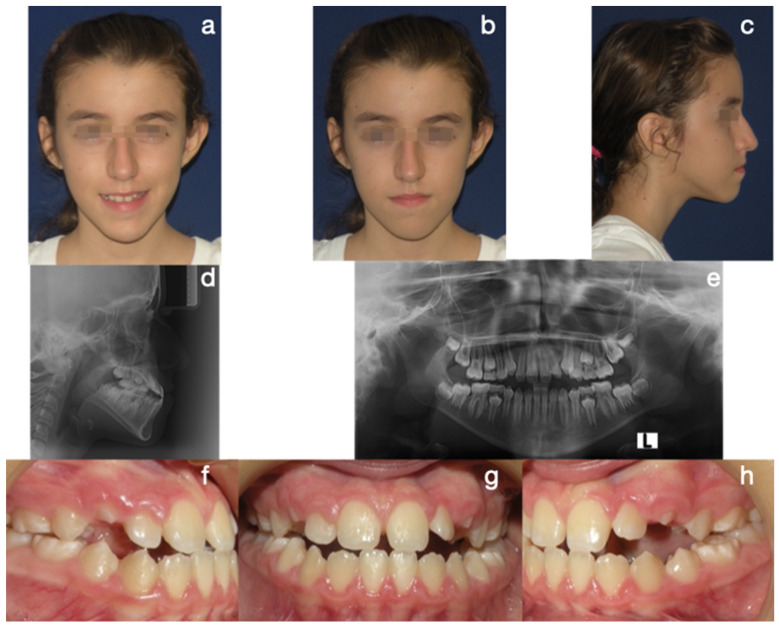
Case 3 corresponds to patient 9 described in Table 1. Protocol C (bilateral condylectomy followed by orthopedics and/or corrective orthodontics without second surgical intervention). Initial photos: (**a**) frontal extraoral photograph, (**b**) frontal smile photograph, (**c**) lateral extraoral photograph, (**d**) lateral skull radiograph, (**e**) panoramic radiograph, (**f**) lateral right intraoral photograph, (**g**) frontal intraoral photograph, (**h**) lateral left intraoral photograph.

**Figure 8 diagnostics-15-00809-f008:**
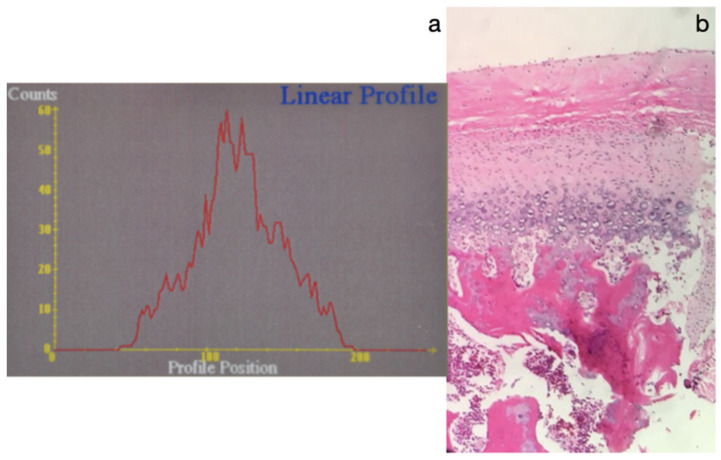
(**a**) Histogram of the SPECT test, with an uptake ratio in the right condyle of 1.45 and left condyle of 1.38. (**b**) Histological slice image with bilateral 0.6 mm thickness in the soft condyle.

**Figure 9 diagnostics-15-00809-f009:**
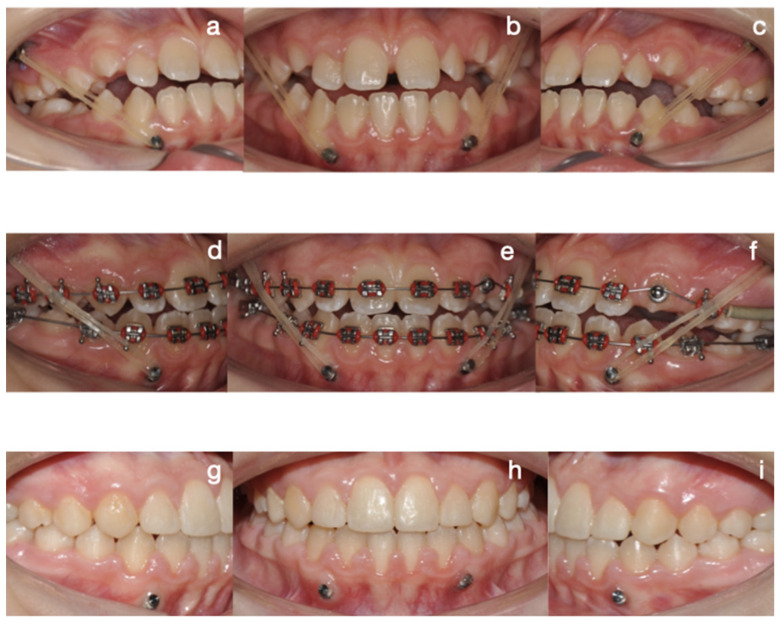
Intraoral treatment sequence: (**a**) right lateral post-surgical photograph, (**b**) frontal post-surgical photograph with skeletal anchorage and elastic mechanics, (**c**) left lateral post-surgical photograph with Class III elastic vector and 250 g force, (**d**) right lateral corrective orthodontic photograph with Class III elastic vector and 250 g force, (**e**) frontal photograph, (**f**) left lateral photograph, (**g**) right lateral final photograph, (**h**) frontal final photograph, (**i**) left lateral final photograph.

**Figure 10 diagnostics-15-00809-f010:**
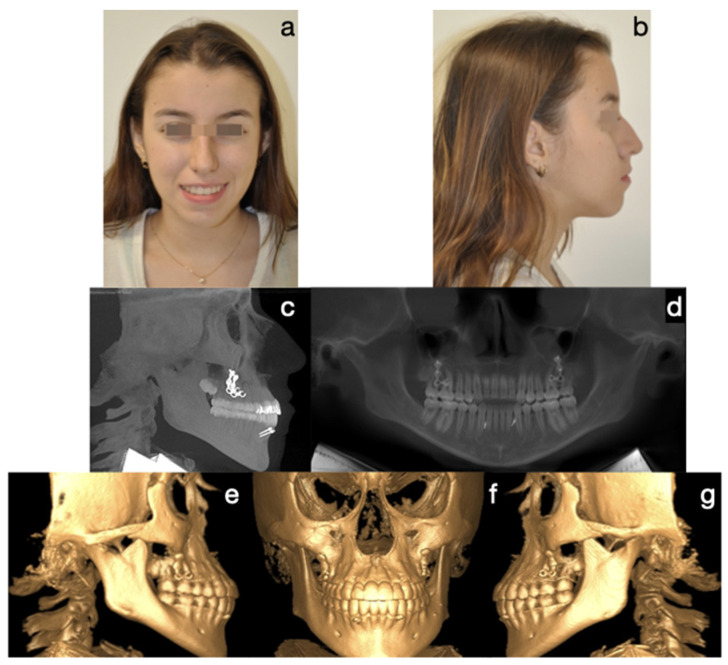
Final photos. Correction of the skeletal discrepancy, positive overjet, and adequate incisor emergence profile. Zygomatic plates and symphyseal screws were kept for the retention stage. (**a**) Frontal extraoral photograph, (**b**) lateral extraoral photograph, (**c**) lateral skull radiograph, (**d**) panoramic radiograph, (**e**) lateral CBCT, (**f**) frontal CBCT, (**g**) left lateral CBCT.

**Table 1 diagnostics-15-00809-t001:** Demographic characteristics, clinical findings, ratio result, surgical procedure, histopathological results, treatment protocol, and post-surgical follow-up.

Patient	Age	Gender	Clinical Highlights	Ratio Results	Magnitude of the Condilectomy	Histopathological Findings	Treatment Type	Follow Up
1	14	F	Severe mandibular prognathism. Elongated condyles. Accelerated mandibular growth after menarche (13 years). Negative overjet of 12 mm. No family history of Class III.	Right condyle: 1.4 Left condyle: 1.4	Right condyle: 6 mm Left condyle: 6 mm	Soft condyle thickness. Right 0.6 mm. Left 0.7 mm. Positive findings for BCH.	Type A. Bilateral condylectomy, orthodontic decompensation and 12 months later bimaxillary orthognathic surgery.	12 months
2	12	F	Severe mandibular prognathism. Elongated condyles. Accelerated mandibular growth after menarche (11 years). Negative overjet of 8 mm No family history of Class III	Right condyle: 1.38 Left condyle: 1.38	Right condyle: 6 mm Left condyle: 6 mm	Soft condyle thickness. Right 0.6 mm. Left 0.6 mm. Positive findings for BCH.	Type C. Bilateral condylectomy, followed by corrective orthodontics.	36 months
3	25	M	Severe mandibular prognathism. Elongated condyles. Accelerated mandibular growth after 15 years of age. Negative overjet of 14 mm. Patient with bicigotic twin presenting unilateral condylar hyperplasia.	Positive report for uptake ratio above normal values in condyles, with respect to clivus. Did not provide value	Right condyle: 5 mm Left condyle: 7 mm	Increased thickness of the layers of the soft condyle, without giving thickness value. Increased proliferation of hypertrophic chondrocytes on medullary bone. Positive findings for BCH.	Type A. Bilateral condylectomy, orthodontic decompensation and 12 months later bimaxillary orthognathic surgery with mentoplasty.	60 monts
4	30	M	Severe mandibular prognathism. Elongated condyles. Accelerated mandibular growth from the age of 15 years and beyond the age of 20 years. Negative overjet of 12 mm. No family history of Class III. Failed orthodontic treatment.	Positive report for uptake ratio above normal values in condyles, with respect to clivus. Did not provide value	Right condyle: 6 mm Left condyle: 6 mm	Increased thickness of the layers of the soft condyle, without giving thickness value. Increased proliferation of hypertrophic chondrocytes on medullary bone. Positive findings for BCH.	Type B. Bilateral condylectomy, bimaxillary orthognathic surgery and mentoplasty all in the same surgical time.	60 months
5	17	M	Severe mandibular prognathism. Elongated condyles. Accelerated mandibular growth since the age of 14 years. Negative overjet of 16 mm. No family history of Class III. Failed orthopedic treatments. Depressive mood episodes.	Positive report for uptake ratio above normal values in condyles, with respect to clivus. Did not provide value	Right condyle: 6 mm Left condyle: 6 mm	Increased thickness of the layers of the soft condyle, without giving thickness value. Increased proliferation of hypertrophic chondrocytes on medullary bone. Positive findings for BCH.	Type B. Bilateral condylectomy, bimaxillary orthognathic surgery and mentoplasty all in the same surgical time.	60 months
6	18	F	Severe mandibular prognathism. Elongated condyles. Accelerated mandibular growth since the age of 11 years. Hypoplasia of the middle third of the face. Negative overjet of 6 mm. No family history of Class III. Failed orthopedic and orthodontic compensation treatment.	Right condyle: 1.36 Left condyle: 1.36	Right condyle: 6 mm Left condyle: 6 mm	Increased thickness of the layers of the soft condyle, without giving thickness value. Increased proliferation of hypertrophic chondrocytes on medullary bone. Positive findings for BCH.	Type A. Bilateral condylectomy, orthodontic decompensation for future bimaxillary orthognathic surgery.	12 months
7	16	M	Severe mandibular prognathism. Elongated condyles and posterior divergence. Accelerated mandibular growth since the age of 14 years. Hypoplasia of the middle third of the face. Negative overjet of 8 mm. No family history of Class III. Low self-perception of facial appearance.	Positive report for uptake ratio above normal values in condyles, with respect to clivus. Did not provide value	Right condyle: 7 mm Left condyle: 5 mm	Increased thickness of the layers of the soft condyle, without giving thickness value. Increased proliferation of hypertrophic chondrocytes on medullary bone. Positive findings for BCH.	Type A. Bilateral condylectomy, orthodontic decompensation for future bimaxillary orthognathic surgery.	24 months
8	17	M	Severe mandibular prognathism. Elongated condyles and posterior divergence. Accelerated mandibular growth since the age of 12 years. Hypoplasia of the middle third of the face. Negative overjet of 8 mm. No family history of Class III. Low self-perception of facial appearance.	Right condyle: 2.25 Left condyle: 2.30	Right condyle: 9 mm Left condyle: 10 mm	Increased thickness of the layers of the soft condyle, without giving thickness value. Increased proliferation of hypertrophic chondrocytes on medullary bone. Positive findings for BCH.	Type C. Bilateral condylectomy, followed by corrective orthodontics of malocclusion.	8 months
9	11	F	Mandibular prognathism. Elongated and posterior divergent condyles. Accelerated mandibular growth after menarche (11 years). Edge-to-edge bite. Incisive compensation. No family history of Class III.	Right condyle: 1.45 Left condyle: 1.38	Right condyle: 5 mm Left condyle: 5 mm	Soft condyle thickness. Right 0.6 mm. Left 0.6 mm. Positive findings for BCH.	Type C. Bilateral condylectomy, followed by skeletally anchored orthopedics, use of elastics with orthopedic forces and corrective orthodontics of the malocclusion.	48 months
10	16	M	Class III since infancy. Severe mandibular prognathism. Elongated condyles. Accelerated growth from the age of 14 years. Negative overjet of 5 mm. Severely retroinclined lower incisors.	Right condyle: 2.1 Left condyle: 2.2	Right condyle: 9 mm Left condyle: 12 mm	Increased thickness of the layers of the soft condyle, without giving thickness value. Increased proliferation of hypertrophic chondrocytes on medullary bone. Positive findings for BCH.	Type A. Bilateral condylectomy, orthodontic decompensation and 9 months later bimaxillary orthognathic surgery.	42 months

F, female; M, male.

## Data Availability

Data are available upon request due to restrictions, such as privacy or ethics.

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
