# Peer review of "Bilateral Condylar Hyperplasia: Importance of Its Diagnosis in the Treatment and Long-Term Stability of Skeletal Class III Correction"

_diagnostics, 2025, doi:10.3390/diagnostics15070809_

Round 1

Reviewer 1 Report

Comments and Suggestions for Authors

Dear Authors,Thank You for a pleasure to read Your article. I have several notes and comments for You to improve Your manuscript. Please, check English. Title

It is too long and does not explain the idea of Your work. Please, re-write it and add the type (a case series).

Abstract

Results are materials and methods description. Please, replace it with results exactly.

Please, check the keywords with MeSH

Introduction

It is too short, please, add some information, for example, about diagnostics and treatment, etiology or other.

Lines 68-72 are more appropriate for other sections as discussion, for example.

Materials and methods

Please, describe the patients’ characteristics here, also, write the inclusion and exclusion criteria.

As You have special consent form in special center, please, attach sample for non-published materials.

Please, explain the algorithm of choosing and creating the surgical plan. Lines 104-105 are not clear.

Line 112. Please, write what exactly orthognathic surgery was.

Write, what patients were in each group (sex, age etc.) besides table 1.

Please, add for table 1 the results of scintigraphy for all cases

Discussion

Please, compare Your results with other study but do not write or re-write here the results.

Also, please, describe one case with details for each group according to Your algorithm for treatment plan without simple photos.

Sincerely, Reviewer

Comments on the Quality of English Language

I have no expertise for English language quality decision

Author Response

Cover Letter

Manuscript ID: diagnostics-3495509

Reviewers, thank you very much for your comments and observations, which have helped us improve the manuscript and obtain a better publication result.

Reviewer 1

Title: It is too long and does not explain the idea of your work. Please, re-write it and add the type (a case series).

Answer: The title was reduced in length, leaving the sense of what the authors want to express, and the type of study was added: case series. All changes are highlighted in yellow.

Abstract: Results are materials and methods description. Please, replace it with results exactly. Please, check the keywords with MeS.

Answer: We reorganized the results paragraph and verified that all keywords were in the MeS descriptors. The changes are highlighted in yellow.

Introduction: It is too short, please, add some information, for example, about diagnostics and treatment, etiology or other. Lines 68-72 are more appropriate for other sections as discussion, for example.

Answer: Following the recommendations, the introduction was lengthened, the information was reorganized and expanded, and the last paragraph was modified. It is highlighted in yellow.

Materials and methods:

Please, describe the patients’ characteristics here, also, write the inclusion and exclusion criteria. As You have special consent form in special center, please, attach sample for non-published materials. Please, explain the algorithm of choosing and creating the surgical plan. Lines 104-105 are not clear.

Line 112. Please, write what exactly orthognathic surgery was.

Write, what patients were in each group (sex, age etc.) besides table 1.

Please, add for table 1 the results of scintigraphy for all cases.

Answer: The exclusion and inclusion criteria for the study were added in the materials and methods. Lines 98 to 103. They are highlighted in yellow.

Following the recommendation, the informed consent form used in each case was added to the unpublished supplementary material.

From line 124, the three treatment proposals are generally proposed. Type A: Bilateral condylectomy, followed by orthodontic decompensation and then orthognathic surgery to correct the sequelae.

Type B: Bilateral condylectomy and orthognathic surgery at the same surgical time. The type of surgery depends on the structures to be corrected. Type C: Bilateral condylectomy and then orthodontics and/or maxillary orthopedics. Only one surgical time.

The order and procedures used in protocol B are described in lines 133 to 136, following the recommendation. They are highlighted in yellow.

Demographic characteristics, clinical findings, Ratio result, surgical procedure, histopathological results, treatment protocol, and post-surgical follow-up are described in table 1. results section

In four of the 10 patients, the nuclear medicine center did not send the quantitative result. It was limited to sending the histogram and the nuclear medicine physician's analysis certifying the radiopharmaceutical uptake in the condyles, above the clivus and with a positive diagnosis of bilateral condylar hypercaptation adjusted for age. This was reported in the results and in Table 1. It is highlighted in yellow.

Discussion: Please, compare Your results with other study but do not write or re-write here the results.

Also, please, describe one case with details for each group according to Your algorithm for treatment plan without simple photos.

Answer: There are no other clinical studies reported in the literature on the diagnosis and treatment of bilateral condylar hyperplasia, neither in case series nor case-controls. Only Wolford's description in 2014, based on his experience, and without an analyzed sample. This was added in the last paragraph of the discussion as one of the limitations of the study.

In the penultimate paragraph of the discussion, the factors that influence the type of procedure to be followed are described. According to the demographic characteristics, the severity of the alteration, the psycho-emotional component and even the health insurance coverage, it is decided which protocol to follow (A, B or C). The article shows a case for each protocol.

Reviewer 2 Report

Comments and Suggestions for Authors

Dear Authors,

thank you for this interesting article of yours. Here are some suggestions how to improve it:

  1. Line 49-50, there should also be a statement in which if we have an early class III due to maxillary hypoplasia, the mandibles overgrowth could be larger. Please, focus on „hypoplasia” in this case, because the condition id 3D.
  2. Line 92-95 - which exactly?
  3. When writing about the patients, please add the information in the results, that the patients were healthy and no congenital syndrome was diagnosed (or was it?)
  4. In the title add the information this is a study, but also case presentation
  5. In the discussion, add the information on possible congenital effects that are combined with condes (eg.doi:10.17219/dmp/186086) - due to that you would develop the asymmetries. Also, mention condylectomy as a possible treatment for those asymmetric (syndromic and non-syndromic) cases
  6. Add the limitations of the study

Summing up, the paper is of high value and I hope for further studies on those cases. Thank you

Author Response

Cover Letter

Manuscript ID: diagnostics-3495509

Reviewers, thank you very much for your comments and observations, which have helped us improve the manuscript and obtain a better publication result.

Reviewer 2

thank you for this interesting article of yours. Here are some suggestions how to improve it:

  1. Line 49-50, there should also be a statement in which if we have an early class III due to maxillary hypoplasia, the mandibles overgrowth could be larger. Please, focus on „hypoplasia” in this case, because the condition id 3D.

Answer: In the first paragraph of the introduction, we focused on condylar hyperplasia and how it produces excessive and disproportionate growth of the condyle, affecting not only the structures that make up the temporomandibular joint, but secondarily and compensatorily the structures such as the maxilla and occlusion. We do not mention maxillary hypoplasia, because it is not influenced by condylar overgrowth. Condylar growth is of the endochondral type, different from that of the maxilla. However, we accept the recommendation and place it in the first paragraph of the discussion. In which we mention the prevalence of maxillary hypoplasia in skeletal Class III.

  1. Line 92-95 - which exactly?

Aswer: The clinical and imaging evaluation, described in these lines, is part of the diagnostic process proposed for these cases. The comparison with reference values related to the population under study, is with the research described in the discussion. Reference 29.

  1. When writing about the patients, please add the information in the results, that the patients were healthy and no congenital syndrome was diagnosed (or was it?)

Answer: The exclusion and inclusion criteria for the study were added in the materials and methods. Lines 98 to 103. They are highlighted in yellow.

  1. In the title add the information this is a study, but also case presentation.

Answer: The type of study was added in the title, case series.

  1. In the discussion, add the information on possible congenital effects that are combined with condes (eg.doi:10.17219/dmp/186086) - due to that you would develop the asymmetries. Also, mention condylectomy as a possible treatment for those asymmetric (syndromic and non-syndromic) cases.

Answer: This reference: “Most common congenital syndromes with facial asymmetry” A narrative review. It is important in asymmetric cases, related to congenital syndromes.

We understand the importance of the topic, but it is far from the research question in the study. Bilateral condylar hyperplasia, related to severe mandibular prognathism.

  1. Add the limitations of the study

Answer: In the last paragraph of the discussion, according to the recommendation, the limitations and the proposal for future studies were included. It is highlighted in yellow.

Summing up, the paper is of high value and I hope for further studies on those cases. Thank you

Round 2

Reviewer 1 Report

Comments and Suggestions for Authors

Dear Authors,

Thank You for manuscript correction.

Introduction now is not too bigger to comparison with previous version. Please, add important information.

For results, please, re-organize groups as You have no all information for all clinical cases or, as I previously asked You, describe one clinical case TOTALLY for each study group according to CARE checklist.

Discussion

You wrote that information about pathology was absent but during simple search in Google Academy I found at least several articles: for example, https://doi.org/10.1016/j.joms.2007.08.046 , Wolford’s article, https://doi.org/10.1016/j.bjoms.2018.07.017 etc.

That is why, please, re-think discussion and re-write it.

Sincerely, Reviewer

Author Response

Cover Letter

Manuscript ID: diagnostics-3495509

Reviewer, thank you very much for your comments and observations

Reviewer 1

Thank You for manuscript correction.

  1. Introduction now is not too bigger to comparison with previous version. Please, add important information.

Answer

Following the recommendation, a paragraph on mandibular prognathism was added. This condition is directly associated with work. Lines 77 to 84. They are highlighted in green.

  1. For results, please, re-organize groups as You have no all information for all clinical cases or, as I previously asked You, describe one clinical case TOTALLY for each study group according to CARE checklist. 

Answer

The recommendation was followed, and in addition to the description made in the explanations of the photos and Table 1, now in the results section, a detailed description of one case for each protocol is described. Everything is highlighted in green in the results section. 

Discussion 

  1. You wrote that information about pathology was absent but during simple search in Google Academy I found at least several articles: for example, https://doi.org/10.1016/j.joms.2007.08.046 , Wolford’s article, https://doi.org/10.1016/j.bjoms.2018.07.017 etc.

Answer

Unilateral Condylar hyperplasia is widely described in the literature, in fact our research group has been working deeply on it for the last 15 years. What we commented in the answer is that bilateral condylar hyperplasia is not even moderately described in the literature. The diagnostic features, the natural history of the disease, the anatomic and metabolic imaging tests, and the treatment schemes are not adequately referenced.

The article you mention by Wolford, as well as the other article by Higginson, deal with unilateral condylar hyperplasia leading to progressive facial asymmetry. Different from this study, which purports to show bilateral condylar hyperplasia leading to severe mandibular prognathism. Two entities in manifestations, diagnosis, and treatment approach that are different. The above is mentioned in the introduction.

 We apologize, but we are unable to address this suggestion because of the above.

Round 3

Reviewer 1 Report

Comments and Suggestions for Authors

Dear Authors,

Thank You for Your correction and answers. 

Sincerely, Reviewer